# A Preliminary Logic-based Approach for Explanation Generation

**Stylianos L. Vasileiou**
Computer Science and Engineering
Washington University in St. Louis
v.stylianos@wustl.edu

**William Yeoh**
Computer Science and Engineering
Washington University in St. Louis
wyeoh@wustl.edu

**Tran Cao Son**
Computer Science
New Mexico State University
tson@cs.nmsu.edu

## Abstract

In an explanation generation problem, an agent needs to identify and explain the reasons for its decisions to another agent. Existing work in this area is mostly confined to planning-based systems that use automated planning approaches to solve the problem. In this paper, we approach this problem from a new perspective, where we propose a general logic-based framework for explanation generation. In particular, given a knowledge base $KB_1$ that entails a formula $\phi$ and a second knowledge base $KB_2$ that does not entail $\phi$, we seek to identify an explanation $\epsilon$ that is a subset of $KB_1$ such that the union of $KB_2$ and $\epsilon$ entails $\phi$. We define two types of explanations, model- and proof-theoretic explanations, and use cost functions to reflect preferences between explanations. Further, we present our algorithm implemented for propositional logic that compute such explanations and empirically evaluate it in random knowledge bases and a planning domain.

## Introduction

With increasing proliferation and integration of AI systems in our daily life, there is a surge of interest in *explainable AI*, which includes the development of AI systems whose actions can be easily understood by humans. Driven by this goal, *machine learning* (ML) researchers have begun to classify commonly used ML algorithms according to different dimensions of explainability (Guidotti *et al.* 2018); improved the explainability of existing ML algorithms (Vaughan *et al.* 2018; Alvarez Melis and Jaakkola 2018; Petkovic *et al.* 2018); as well as proposed new ML algorithms that trade off accuracy for increasing explainability (Dong *et al.* 2017; Gilpin *et al.* 2018).[1]

In contrast, researchers in the *automated planning* community have mostly taken a complementary approach. While there is some work on adapting planning algorithms to find easily explainable plans[2] (i.e., plans that are easily understood and accepted by a human user) (Zhang *et al.* 2017), most work has focused on the *explanation generation problem* (i.e., the problem of identifying explanations of plans found by planning agents that when presented to users, will allow them to understand and accept the proposed plan) (Langley 2016; Kambhampati 1990). Within this context, researchers have tackled the problem where the model of the human user may be (1) inconsistent with the model of the planning agent (Chakraborti *et al.* 2017b); (2) must be learned (Zhang *et al.* 2017); and (3) a different form or abstraction than that of the planning agent (Sreedharan *et al.* 2018; Tian *et al.* 2016). However, a common thread across most of these works is that they, not surprisingly, employ mostly automated planning approaches. For example, they often assume that the models of both the agent and human are encoded in PDDL format.

In this paper, we approach the explanation generation problem from a different perspective – one based on *knowledge representation and reasoning* (KR). We propose a general logic-based framework for explanation generation, where given a knowledge base $KB_1$ (of an agent) that entails a formula $\phi$ and a knowledge base $KB_2$ (of a human user) that does not entail $\phi$, the goal is to identify an explanation $\epsilon \subseteq KB_1$ such that $KB_2 \cup \epsilon$ entails $\phi$. We define two types of explanations, model- and proof-theoretic explanations, and use cost functions to reflect preferences between explanations. Further, we present an algorithm, implemented for propositional logic, that computes such explanations and evaluate its performance experimentally in random knowledge bases as well as in a planning domain.

In addition to providing an alternative approach to

---

[1]While the term *interpretability* is more commonly used in the ML literature and can be used interchangeably with *explainability*, we use the latter term as it is more commonly used broadly across different subareas of AI.

---

[2]Also called *explicable* plans in the planning literature.

solve the same explanation generation problem tackled thus far by the automated planning community, our approach has the merit of being more generalizable to other problems beyond planning problems as long as they can be modeled using a logical KR language.

# Preliminaries

## Logic

A *logic* $L$ is a tuple $(KB_L, BS_L, ACC_L)$ where $KB_L$ is the set of well-formed knowledge bases (or theories) of $L$ – each being a set of formulae. $BS_L$ is the set of possible belief sets; each element of $BS_L$ is a set of syntactic elements representing the beliefs $L$ may adopt. $ACC_L : KB_L \rightarrow 2^{BS_L}$ describes the *"semantics"* of $L$ by assigning to each element of $KB_L$ a set of acceptable sets of beliefs. For each $KB \in KB_L$ and $B \in ACC_L(KB)$, we say that $B$ is a *model* of $KB$. A logic is monotonic if $KB \subseteq KB'$ implies $ACC_L(KB') \subseteq ACC_L(KB)$.

**Example 1** *Assume that $L$ refers to the propositional logic over an alphabet $P$. Then, $KB_L$ is the set of propositional theories over $P$, $BS_L = 2^P$, and $ACC_L$ maps each theory $KB$ into the set of its models in the usual sense.*

We say that a $KB$ is *consistent* if $ACC_L(KB) \neq \emptyset$. A formula $\varphi$ in the logic $L$ is *entailed* by $KB$, denoted by $KB \models_L \varphi$, if $ACC_L(KB) \neq \emptyset$ and $\varphi \in B$ for every $B \in ACC_L(KB)$.

For our later use, we will assume that a negation operator $\neg$ over formulas exists; and $\varphi$ and $\neg\varphi$ are contradictory with each other in the sense that for any $KB$ and $B \in ACC_L(KB)$, if $\varphi \in B$ then $\neg\varphi \notin B$; and if $\neg\varphi \in B$ then $\varphi \notin B$. $\epsilon \subseteq KB$ is called a *sub-theory* of $KB$. A theory $KB$ *subsumes* a theory $KB'$, denoted by $KB \lhd KB'$ if $ACC_L(KB) \subset ACC_L(KB')$.

Conclusions of a knowledge base can also be derived using rules. A rule system $\Sigma_L$ of a logic $L$ is a set of rules of the form

$$\varphi_1, \ldots, \varphi_k \vdash_L \varphi_0 \qquad (1)$$

where $\varphi_i$ are formulas. The left hand side could be empty. For a rule $r$ of the form (1), $body(r)$ (resp. $head(r)$) denotes the left (resp. right) side of $r$. Intuitively, a rule $r$ states that if the body is true then the head is also true.

Given a knowledge base $KB$ and a rule system $\Sigma_L$, we say $KB \vdash_{\Sigma_L} \varphi$ if either $\varphi \in KB$ or there exists a sequence of rules $r_1, \ldots, r_n$ in $\Sigma_L$ such that $body(r_1) \subseteq KB$, $head(r_n) = \varphi$, $head(r_i) \in KB$ for $i = 1, \ldots, n-1$, and $body(r_i) \subseteq KB$ or $body(r_i) \subseteq body(r_1) \cup \{head(r_j) \mid j = 1, \ldots, i-1\}$ for every $i = 2, \ldots, n$. We call the sequence $\epsilon = \langle r_1; \ldots; r_n \rangle$ as a *proof* from $KB$ for $\varphi$ w.r.t. $\Sigma_L$ and say that the proof $\epsilon$ has the length $n$.

$\Sigma_L$ is said to be *sound* if for every $\varphi$, $KB \vdash_{\Sigma_L} \varphi$ implies $KB \models_L \varphi$. It is *complete* if for every $\varphi$, $KB \models_L \varphi$ implies $KB \vdash_{\Sigma_L} \varphi$.

## Classical Planning as Boolean Satisfiability

A classical planning problem (Russell and Norvig 2009) can be naturally encoded as an instance of propositional satisfiability (Kautz *et al.* 1992). The basic idea is the following: Given a planning problem $P$, find a solution for $P$ of length $n$ by creating a propositional formula that represent the initial state, goal, and the action dynamics for $n$ time steps. This is referred to as the *bounded planning problem* $(P, n)$, and we define the formula for $(P, n)$ such that: *any* model of the formula represents a solution to $(P, n)$ and if $(P, n)$ has a solution, then the formula is satisfiable.

We encode $(P, n)$ as a formula $\Phi$ such that $\langle a_0, a_1, \ldots, a_{n-1} \rangle$ is a solution for $(P, n)$ if and only if $\Phi$ can be satisfied in a way that makes the fluents $a_0, a_1, \ldots, a_{n-1}$ true. The formula $\Phi$ is a conjunction of the following formulae:

- **Initial State:** Let $F$ be the set of possible facts in the planning problem:

$$\bigwedge \{f_0 | f \in s_0\} \wedge \bigwedge \{\neg f_0 | f \in F \setminus \{s_0\}\}$$

- **Goal:** Let $G$ be the set of goal facts:

$$\bigwedge \{f_n | f \in G\}$$

- **Action Scheme:** For every action $a_i$ at time step $i$:

$$a_i \Rightarrow \bigwedge \{f_i | f \in \text{Precondition(a)}\}$$
$$a_{i+1} \Rightarrow \bigwedge \{f_{i+1} | f \in \text{Add(a)}\}$$
$$a_{i+1} \Rightarrow \bigwedge \{\neg f_{i+1} | f \in \text{Delete(a)}\}$$

- **Explanatory Frame Axioms:** Formulae describing what does not change between steps $i$ and $i + 1$:

$$\neg f_i \wedge f_{i+1} \Rightarrow \bigvee \{a_i | f \in \text{ADD(a)}\}$$
$$f_i \wedge \neg f_{i+1} \Rightarrow \bigvee \{a_i | f \in \text{DEL(a)}\}$$

- **Complete Exclusion Axioms:** Only one action can occur at each time step:

$$\neg a_i \vee \neg b_i$$

Finally, we can *extract* a plan by finding an assignment of truth values that satisfies $\Phi$ (i.e., for $i = 0, \ldots, n-1$, there will be exactly one action $a$ such that $a_i = True$). This could be easily done by using a satisfiability algorithm, such as the well-known DPLL algorithm (Davis *et al.* 1962).

In this paper, we will mostly use examples from propositional logic. We make use of the fact that the resolution rule is sound and complete in first-order logic (Robinson 1965), and hence, in propositional logic. This allows us to utilize the DPLL algorithm in computing proofs for a formula given a knowledge base.

## Two Accounts of Explanations

In this section, we introduce the notion of an explanation in the following setting:

> **Explanation Generation Problem:** Given two knowledge bases $KB_1$ and $KB_2$ and a formula $\varphi$ in a logic $L$. Assume that $KB_1 \models_L \varphi$ and $KB_2 \not\models_L \varphi$. The goal is to identify an explanation (i.e., a set of formulas) $\epsilon \subseteq KB_1$ such that $KB_2 \cup \epsilon \models \varphi$.

We first define the notion of a support of a formula w.r.t. a knowledge base.

**Definition 1 (Support)** *Assume that $KB \models_L \varphi$. We say that $\epsilon \subseteq KB$ is a* support *of $\varphi$ w.r.t. $KB$ if $\epsilon \models_L \varphi$. Assume that $\epsilon$ is a support of $\varphi$ w.r.t. $KB$. We say that $\epsilon \subseteq KB$ is a $\subseteq$-minimal support of $\varphi$ if no proper subtheory of $\epsilon$ is a support of $\varphi$. Furthermore, $\epsilon$ is a $\lhd$-general support of $\varphi$ if there is no support $\epsilon'$ of $\varphi$ w.r.t. $KB$ such that $\epsilon$ subsumes $\epsilon'$.*

We now define below two types of explanations – model-theoretic and proof-theoretic explanations.

### Model-Theoretic Explanations

**Definition 2 ($m$-Explanation)** *Given two knowledge bases $KB_1$ and $KB_2$ in logic $L$ and a formula $\varphi$. Assume that $KB_1 \models_L \varphi$ and $KB_2 \not\models_L \varphi$.*

*A model-theoretic explanation (or $m$-explanation) for $\varphi$ from $KB_1$ for $KB_2$ is a support $\epsilon$ w.r.t. $KB_1$ for $\varphi$ such that $KB_2 \cup \epsilon \models_L \varphi$.*

**Example 2** *Consider proposition logic theories over the set of propositions $\{a, b, c\}$ with the usual definition of models, satisfaction, etc. Assume $KB_1 = \{a, b, a \rightarrow c, a \wedge b \rightarrow c\}$ and $KB_2 = \{a\}$. We have that $\epsilon_1 = \{a, a \rightarrow c\}$ and $\epsilon_2 = \{a, b, a \wedge b \rightarrow c\}$ are two $\subseteq$-minimal supports of $c$ w.r.t. $KB_1$. Only $\epsilon_1$ is a $\lhd$-general support of $c$ w.r.t. $KB_1$ since $\epsilon_2 \lhd \epsilon_1$.*

*Both $\epsilon_1$ and $\epsilon_2$ can serve as $m$-explanations for $c$ from $KB_1$ for $KB_2$. Of course, $KB_1$ is itself an $m$-explanation for $c$ from $KB_1$ for $KB_2$.*

*Consider $KB_3 = \{a, \neg b\}$. In this case, we have that only $\epsilon_1$ is an $m$-explanation for $c$ from $KB_1$ for $KB_3$.*

*Now, consider $KB_4 = \{\neg a\}$. In this case, we have no $m$-explanation for $c$ from $KB_1$ for $KB_4$.*

**Proposition 1** *For two knowledge bases $KB_1$ and $KB_2$ in a monotonic logic $L$, if $KB_1 \models_L \varphi$ and $KB_2 \models_L \neg\varphi$, then there exists no $m$-explanation for $\varphi$ from $KB_1$ for $KB_2$.*

The $KB_4$ in Example 2 and Proposition 1 show that $m$-explanations alone might be insufficient. Sometimes, we also need to persuade the other agent that its knowledge base is not correct. We leave this for the future. In this paper, we assume that $KB_2 \not\models_L \neg\varphi$ and $KB_2 \not\models_L \varphi$ and, thus, an $m$-explanation always exists.

## Proof-Theoretic Explanations

**Definition 3 ($p$-explanation)** *Given a logic $L$ with a sound and complete rule system $\Sigma_L$ and two knowledge bases $KB_1$ and $KB_2$ in logic $L$ and a formula $\varphi$. Assume that $KB_1 \vdash_L \varphi$ and $KB_2 \not\vdash_L \varphi$.*

*A proof-theoretic explanation (or $p$-explanation) for $\varphi$ from $KB_1$ for $KB_2$ is a proof $\langle r_1; \ldots; r_n \rangle$ from $KB_1$ for $\varphi$ such that $KB_2 \cup (\bigcup_{i=1}^{n} body(r_i) \cap KB_1) \vdash_{\Sigma_L} \varphi$ and $KB_2 \cup (\bigcup_{i=1}^{n} body(r_i) \cap KB_1)$ is consistent.*

**Example 3** *Consider the theories $KB_1 = \{a, b, a \rightarrow c, a \wedge b \rightarrow c\}$ and $KB_2 = \{a\}$ from Example 2. Let us assume that $\Sigma_L$ is the set of rules of the form $l \vdash_L l$ and $l, \neg l \vee p \vdash p$ for any literals $l, p$ in the language of $KB_1$ and $KB_2$. Then, $\langle a, \neg a \vee c \vdash_L c \rangle$ is a proof from $KB_1$ for $c$, which is also a $p$-explanation for $\varphi$ from $KB_1$ for $KB_2$.*

*Likewise, $\langle a \vdash_L a; b \vdash_L b; a, \neg a \vee \neg b \vee c \vdash_L \neg b \vee c; b, \neg b \vee c \vdash_L c \rangle$ is a $p$-explanation for $c$ from $KB_1$ for $KB_2$.*

**Proposition 2** *Assume that $\Sigma_L$ is a sound and complete rule system of a logic $L$, $KB_1$ is a knowledge base, and $\varphi$ is a formula in $L$. For each proof $\langle r_1; \ldots; r_n \rangle$ from $KB_1$ for $\varphi$ w.r.t. $\Sigma_L$, $\bigcup_{i=1}^{n} body(r_i) \cap KB_1$ is a support of $\varphi$ w.r.t. $KB_1$.*

Proposition 2 implies that each proof from $KB_1$ for $\varphi$ could be identified as a $p$-explanation for $\varphi$ from $KB_1$ if $\Sigma_L$ is sound and complete. This provides the following relationship between $m$-explanations and $p$-explanations.

**Proposition 3** *Assume that $\Sigma_L$ is a sound and complete rule system of a logic $L$, $KB_1$ and $KB_2$ are two knowledge bases in $L$, and $\varphi$ is a formula in $L$. Then,*

- *for each $m$-explanation $\epsilon$ for $\varphi$ from $KB_1$ for $KB_2$, there exists a $p$-explanation $\langle r_1; \ldots; r_n \rangle$ for $\varphi$ from $KB_1$ for $KB_2$ such that $\bigcup_{i=1}^{n} body(r_i) \cap KB_1 \subseteq \epsilon$; and*

- *for each $p$-explanation $\langle r_1; \ldots; r_n \rangle$ for $\varphi$ from $KB_1$ for $KB_2$, $\bigcup_{i=1}^{n} body(r_i) \cap KB_1$ is an $m$-explanation for $\varphi$ from $KB_1$ for $KB_2$.*

## Preferred Explanations

Given $KB_1$ and $KB_2$ and a formula $\varphi$, there might be several ($m$- or $p$-) explanations for $\varphi$ from $KB_1$ for $KB_2$. For brevity, we will now use the term $x$-explanation for $x \in \{m, p\}$ to refer to an $x$-explanation for $\varphi$ from $KB_1$ for $KB_2$. Obviously, not all explanations are equal. One might preferred a subset minimal $m$-explanation or a shortest length $p$-explanation over others. We will next define a general preferred relation among explanations.

We assume a cost function $\mathcal{C}_L^x$ that maps pairs of knowledge bases and sets of explanations to non-negative real values, i.e.,

$$\mathcal{C}_L^x : KB_L \times \Omega \rightarrow \mathcal{R}^{\geq 0} \tag{2}$$

where $\Omega$ is the set of $x$-explanations and $\mathcal{R}^{\geq 0}$ denotes the set of non-negative real numbers. Intuitively, this function can be used to characterize different complexity measurements of an explanation.

A cost function $\mathcal{C}_L^m$ is *monotonic* if for any two $m$-explanations $\epsilon_1 \subseteq \epsilon_2$, $\mathcal{C}_L^m(KB, \epsilon_1) \leq \mathcal{C}_L^m(KB, \epsilon_2)$. A cost function $\mathcal{C}_L^p$ is *monotonic* if for any two $p$-explanations $\epsilon_1$ and $\epsilon_2$ such that $\epsilon_1$ is a subsequence of $\epsilon_2$, $\mathcal{C}_L^p(KB, \epsilon_1) \leq \mathcal{C}_L^m(KB, \epsilon_2)$.

$\mathcal{C}_L^x$ induces a preference relation $\prec_{KB}$ over explanations as follows.

**Definition 4 (Preferred Explanation)** *Given a cost function $\mathcal{C}_L^x$, a knowledge base $KB_2$, and two $x$-explanations $\epsilon_1$ and $\epsilon_2$ for $KB_2$, we say $\epsilon_1$ is* preferred *over $\epsilon_2$ w.r.t. $KB_2$ (denoted by $\epsilon_1 \preceq_{KB_2}^x \epsilon_2$) iff*

$$\mathcal{C}_L^x(KB_2, \epsilon_1) \leq \mathcal{C}_L^x(KB_2, \epsilon_2) \tag{3}$$

*and $\epsilon_1$ is* strictly preferred *over $\epsilon_2$ w.r.t. $KB_2$ (denoted by $\epsilon_1 \prec_{KB_2}^x \epsilon_2$) if*

$$\mathcal{C}_L^x(KB_2, \epsilon_1) < \mathcal{C}_L^x(KB_2, \epsilon_2) \tag{4}$$

This allows us to compare explanations as follows.

**Definition 5 (Most Preferred Explanation)** *Given a cost function $\mathcal{C}_L^x$ and a knowledge base $KB_2$, an explanation $\epsilon$ is a* most preferred *$x$-explanation w.r.t. $KB_2$ if there exists no other explanation $\epsilon'$ such that $\epsilon' \prec_{KB_2}^x \epsilon$.*

**Proposition 4** *If $\mathcal{C}_L^x$ is monotonic then the relation $\preceq_{KB_2}^x$ over $x$-explanations is transitive, anti-symmetric, and reflexive; and the relation $\prec_{KB_2}^x$ over $x$-explanations is transitive and anti-symmetric.*

There are several natural monotonic cost functions. Examples for cost functions for $m$-explanations include:

- $c_L^1(KB_2, \epsilon) = |\epsilon|$, the cardinality of $\epsilon$, indicates the number of formulas that need to be explained;

- $c_L^2(KB_2, \epsilon) = |\epsilon \setminus KB_2|$, the cardinality of $\epsilon \setminus KB_2$, indicates the number of *new* formulas that need to be explained;

- $c_L^3(KB_2, \epsilon) = |new\_vars(KB_2, \epsilon)|$ indicates the number of *new* symbols occurring in $\epsilon$ that are not in $KB_2$ and need to be explained;

- $c_L^4(KB_2, \epsilon) = length(\epsilon)$ indicates the number of literals in $\epsilon$ that need to be explained.

Naturally, some of these cost functions can also be combined (e.g., $c_L^2 + c_L^3$ will measure the number of new formulas and new symbols that must be explained).

Observe that the three functions $c_L^1$ and $c_L^4$ are independent from $KB_2$ while $c_L^2$ and $c_L^3$ depend on $KB_2$. A potential advantage of a cost function that is independent from $KB_2$ is that it helps simplify the computation of most preferred explanations.

**Example 4** *Continuing with Example 2, if we use $c_L^1$ as the cost function, then we have that $\epsilon_1 \prec_{KB_2}^m \epsilon_2 \prec_{KB_2}^m KB_1$. Furthermore, $\epsilon_1$ is the most preferred $m$-explanation from $KB_1$ to $KB_2$.*

---

**Algorithm 1:** $\texttt{genExp}(KB_1, KB_2, \varphi)$

**Input:** Logic $L$, formula $\varphi$, KBs $KB_1$ and $KB_2$, cost function $\mathcal{C}_L^x$

**Output:** A most preferred $x$-explanation w.r.t. $\mathcal{C}_L^x$ from $KB_1$ to $KB_2$ for $\varphi$; or $nil$

1   **if** $KB_1 \not\models_L \varphi$ *or* $KB_2 \models_L \varphi$ **then**
2     **return** $nil$

3   **if** $KB_1 \models_L \varphi$ *and* $KB_2 \not\models_L \neg\varphi$ **then**
4     $\epsilon = most\_preferred(KB_1, KB_2, \varphi)$
5     **return** $\epsilon$

---

**Algorithm 2:** $\texttt{most\_preferred}(KB_1, KB_2, \varphi)$

**Input:** Logic $L$, formula $\varphi$, KBs $KB_1$ and $KB_2$, cost function $\mathcal{C}_L^x$

**Output:** A most-preferred explanation w.r.t. $\mathcal{C}_L^x$ from $KB_1$ to $KB_2$ for $\varphi$; or $nil$

1   **repeat**
2     non-deterministically select a potential $x$-explanation $\epsilon$, a minimal element w.r.t. $\mathcal{C}_L^x$ and $KB_2$
3     **if** $\epsilon \models \varphi$ *and* $KB_2 \cup \epsilon \models \varphi$ **then**
4       **return** $\epsilon$
5   **until** *all possible explanations are considered*
6   **return** $nil$

---

## Computing Preferred Explanations

At a high level, Algorithms 1 and 2 can be used for computing most-preferred explanations given a formula $\varphi$ and two knowledge bases $KB_1$ and $KB_2$ of a logic $L$ with the cost function $\mathcal{C}_L^x$. We assume that when computing for $p$-explanations, a sound and complete rule system is available. Our algorithms rely on the existence of an algorithm for checking entailment between knowledge bases and formulas (Lines 1 and 3 in Algorithm 1 and Line 4 in Algorithm 2) and an algorithm for computing a potential explanation that is minimal with respect to a cost function and a knowledge base (Lines 2-3 in Algorithm 2). These two algorithms depend on the logic $L$ and the cost function $\mathcal{C}_L^x$ and need to be implemented for specific logic $L$ and function $\mathcal{C}_L^x$.

In the rest of this section, we discuss the implementation of our algorithms for propositional logic and different cost functions. With propositional logic, it is easy to see that checking for entailment can be done by a SAT solver (e.g., $\texttt{MiniSat}$ (Eén and Sörensson 2003)). We next discuss two algorithm implementations, one for $m$-explanations and one for $p$-explanations, that find an explanation that is minimal with respect to a cost function and a knowledge base.

### Most-Preferred $m$-Explanations

Given a cost function $\mathcal{C}_L^m$ such as $c_L^1$, $c_L^2$, $c_L^3$, or $c_L^4$ as defined in Section , Algorithm 3 computes a most pre-

**Algorithm 3:** most_preferred_m($KB_1, KB_2, \varphi$)

**Input:** Formula $\varphi$, KBs $KB_1$ and $KB_2$, cost function $\mathcal{C}_L^m$

**Output:** A most-preferred $m$-explanation w.r.t. $\mathcal{C}_L^m$ from $KB_1$ to $KB_2$ for $\varphi$; or $nil$

1   $q = [\emptyset]$      % a priority queue of potential explanations

2   $checked = \emptyset$    % a set of sets of elements in $KB_1$ that have been considered

3   **repeat**

4      $\epsilon = dequeue(q)$

5      insert $\epsilon$ into $checked$

6      **if** $\epsilon \models \varphi$ and $KB_2 \cup \epsilon \models \varphi$ **then**

7         **return** $\epsilon$

8      **else**

9         **for** $a \in KB_1$ **do**

10           **if** $\epsilon \cup \{a\} \notin checked$ **then**

11             $v = \mathcal{C}_L^m(KB_2, \epsilon \cup \{a\})$

12             $q = enqueue(\epsilon \cup \{a\})$ % use $v$ as key

13   **until** $q$ is empty

14   **return** $nil$

---

**Algorithm 4:** most_preferred_p($KB_1, KB_2, \varphi$)

**Input:** Formula $\varphi$, KBs $KB_1$ and $KB_2$, cost function $\mathcal{C}_L^p$

**Output:** A most-preferred $p$-explanation w.r.t. $\mathcal{C}_L^p$ from $KB_1$ to $KB_2$ for $\varphi$; or $nil$

1   $q = [\emptyset]$    % priority queue of potential explanations

2   **for** $\epsilon$ in $KB_1$ **do**

3      $v = \mathcal{C}_L^p(KB_2, \epsilon)$

4      $q = enqueue(\langle \epsilon \rangle)$      % use $v$ as key

5   $\Omega = \{\langle \epsilon \rangle \mid \epsilon \in KB_1\}$

6   $checked = \emptyset$

7   **repeat**

8      $\langle \epsilon \rangle = dequeue(q)$

9      insert $(b(\epsilon), c(\epsilon))$ into $checked$

10      **if** $c(\epsilon) = \varphi$ and $KB_2 \cup (b(\epsilon) \cap KB_1) \models \varphi$ **then**

11         **return** $\epsilon$

12      **for** $\epsilon'$ in $\Omega$ **do**

13         **if** $c(\epsilon)$ and $c(\epsilon')$ contain complementary literals and $\frac{c(\epsilon), c(\epsilon')}{\phi}$ holds **then**

14           $\hat{\epsilon} = \langle \epsilon \circ \epsilon'; \frac{c(\epsilon), c(\epsilon')}{\phi} \rangle$

15           **if** $(b(\hat{\epsilon}), \phi) \notin checked$ **then**

16             $v = \mathcal{C}_L^p(KB_2, \hat{\epsilon})$

17             $q = enqueue(\langle \hat{\epsilon} \rangle)$    % use $v$ as key

18   **until** $q$ is empty

19   **return** $nil$

---

ferred $m$-explanations w.r.t. $\mathcal{C}_L^m$ from $KB_1$ to $KB_2$ for $\varphi$ or returns $nil$ if none exists.

The key data structures in the algorithm is a priority queue $q$, initialized to only include the empty set, of potential explanations ordered by their costs (Line 1) and a set $checked$ of invalid explanations that have been considered thus far (line 2). The algorithm repeatedly loops the following steps: (*i*) move the explanation with the smallest cost from the priority queue $q$ to $checked$ (Lines 4-5); (*ii*) check if it is a valid $m$-explanation and return if it is (Lines 6-7); (*iii*) if not, extend the explanation by 1 (with each clause from $KB_1$) and insert the extended explanations into the priority queue $q$ (Lines 8-12). If all potential explanations are exhausted, which means that there are no valid $m$-explanations, then the algorithm returns $nil$ (Line 14). It is straightforward to see that the following proposition holds.

**Proposition 5** *For two propositional theories $KB_1$ and $KB_2$ and a formula $\varphi$, Algorithm 3 returns a most preferred $m$-explanation w.r.t. $\mathcal{C}_L^m$ for $\varphi$ from $KB_1$ to $KB_2$ if one exists.*

## Most-Preferred $p$-Explanations

Given a cost function $\mathcal{C}_L^p$ on $p$-explanations, Algorithm 4 computes a most-preferred $p$-explanation w.r.t. $\mathcal{C}_L^p$ from $KB_1$ to $KB_2$ for $\varphi$ or returns $nil$ if none exists.

We use the following notations in the pseudocode: For a proof $\langle \epsilon \rangle$, where $\epsilon$ is the sequence $\langle r_1; \ldots; r_n \rangle$, we write $c(\epsilon) = head(r_n)$ and $b(\epsilon) = \bigcup_{i=1}^{n} body(r_i)$. We also write $\frac{\varphi_1, \varphi_2}{\varphi}$ to indicate that $\varphi$ is the result of

applying the resolution rule on $\varphi_1$ and $\varphi_2$. And we use $\circ$ to denote the concatenation of two sequences.

The algorithm uses the same two data structures – priority queue $q$ and set $checked$ – as in Algorithm 3. The algorithm first populates the queue $q$ with single-rule proofs consist of single clauses in $KB_1$ (Lines 2-4). Then, it repeatedly loops the following steps: (*i*) move the proof with the smallest cost from the priority queue $q$ to $checked$ (Lines 8-9); (*ii*) check if it is a valid $p$-explanation and return if it is (Lines 10-11); (*iii*) if not, extend the proof by 1 and insert the extended proofs into the priority queue $q$ (Lines 12-17). If all potential proofs are exhausted, which means that there are no valid $p$-explanations, then the algorithm returns $nil$ (Line 19). It is straightforward to see that the following proposition holds.

**Proposition 6** *For two propositional theories $KB_1$ and $KB_2$ and a formula $\varphi$, Algorithm 4 returns a most preferred $p$-explanation w.r.t. $\mathcal{C}_L^p$ for $\varphi$ from $KB_1$ to $KB_2$ if one exists.*

## Plan Explanation Generation

As presented in the preliminaries, we can model a planning problem using the propositional logic language and thus utilize the proposed framework to generate explanations. Particularly, we form the knowledge base of

(a) Experimental Results on Random Knowledge Bases

| $|KB_1|$ | $c_L^1$ | | $c_L^2$ | | $c_L^3$ | | $c_L^4$ | |
|---|---|---|---|---|---|---|---|---|
| | cost | time | cost | time | cost | time | cost | time |
| 20 | 7 | 23ms | 5 | 25ms | 2.4 | 26ms | 14 | 24ms |
| 100 | 15 | 2.5s | 10 | 3.0s | 3.8 | 3.1s | 30 | 2.9s |
| 1000 | 117 | 27m | 97 | 30m | 38 | 32m | 347 | 27m |

(b) Experimental Results on BLOCKSWORLD Domain

| $|KB_1|$ | $c_L^1$ | | $c_L^2$ | | $c_L^3$ | | $c_L^4$ | |
|---|---|---|---|---|---|---|---|---|
| | cost | time | cost | time | cost | time | cost | time |
| 225 | 4 | 15.0s | 1 | 16.0s | 0.5 | 15.5s | 7 | 15.0s |
| 387 | 16 | 2.0m | 12 | 2.2m | 0.5 | 2.2m | 35 | 2.0m |

Table 1: Experimental Results

the agent, namely $KB$, by adding the encoded formula $\Phi$ (represented in CNF clauses) as well as the optimal plan of the specific planning problem. Then, we define the explanation in terms of $KB$ and plan optimality as follows:

**Definition 6 (Optimal Plan Explanation)**
*Given a knowledge base $KB$ and a plan $\pi_n = \langle a_0, a_1, \ldots, a_{n-1} \rangle$, we say that $\pi_n$ is optimal in $KB$ if and only if $KB \models \phi$, where $\forall t = 1, \ldots n-1 : \phi = \neg goal_t$.*

In other words, the formula $\phi$ that we seek to explain is that no plan of lengths 1 to $n-1$ exists, and that a plan of length $n$ exists. Therefore, combined, that plan must be an optimal plan. Now, given a second knowledge base $KB_2$ (i.e that of a human user), where $KB_2 \not\models \phi$, we can compute a model- or proof-theoretic explanation as defined in Definitions 2 and 3.

## Experimental Results

We empirically evaluate our implementation of Algorithm 3 to find $m$-explanations on two synthetically generated benchmarks – random knowledge bases and a planning domain called BLOCKSWORLD – both encoded in propositional logic.[3] We evaluated our algorithm using the four cost functions described in Section . Our algorithm was implemented in Python and experiments were performed on a machine with an Intel i7 2.6GHz processor and 16GB of RAM. We report both the cost of the optimal $m$-explanation found as well as the runtime of the algorithm.

### Random Knowledge Bases

We first evaluated our algorithm on random knowledge bases with clauses in Horn form, where we varied the

---

[3]For random knowledge bases, we used an optimized version that uses a version of backward chaining that finds the set of all possible explanations. This approach works only when the clauses in the knowledge base are in Horn form and is sound and complete for such a case (Russell and Norvig 2009).

cardinality of $KB_1$ (the KB of the agent providing the explanation) from 20 to 1000. To construct $KB_2$ (the KB of the agent receiving the explanation), we randomly chose 25% of the clauses from $KB_1$.

To construct each $KB_1$, we first generated $\frac{|KB_1|}{2}$ random symbols, which will be used in the KB. Then, we iteratively generated clauses of increasing length $l$ from 2 to 7. For each length $l$, we generated $\lfloor \frac{|KB_1|}{2 \cdot l} \rfloor$ clauses using the symbols we previously generated such that each symbol is used at most once in these clauses of length $l$. Each clause is a conjunction of $l-1$ elements as the premise and the final $l^{\text{th}}$ element as the conclusion. For example, a KB with a cardinality of 20, 10 symbols are first generated. Then, 5 clauses of length 2, 3 clauses of length 3, 2 clauses of lengths 4 and 5, and 1 clause of lengths 6 and 7 are generated. Finally, to complete the KB, we add all the symbols that are exclusively in the premise of the clauses generated as facts in the KB. The formula $\varphi$ that we seek to explain is one of the randomly chosen conclusions in the clauses generated, which we ensure is entailed by $KB_1$.

Table 1(a) tabulates our results. We make the following observations:

- As expected, the runtimes increase as $|KB_1|$ increases since the algorithm will need to search over a larger search space.
- As expected, the costs of explanations also increase as $|KB_1|$ increases since the explanations are presumably longer and more complex.
- Finally, the runtimes for cost functions $c_L^1$ and $c_L^4$ are smaller than that of $c_L^2$ and $c_L^3$. The reason is the computation of the costs of possible explanations is faster with the former two cost functions since they are not dependent on $KB_2$ while the computation for the latter two cost functions are dependent on $KB_2$.

### Planning Domain

As we were motivated by the explanation generation problem studied in the automated planning community, we also conducted experiments on BLOCKSWORLD, a planning domain where multiple blocks must be stacked in a particular order on a table.[4]

For these planning problems, we first used FAST-DOWNWARD (Helmert 2006) to find optimal solutions to the planning problem. Then, we translate the planning problem into a SAT problem with horizon $h$ (Kautz *et al.* 1992), where $h$ is the length of the optimal plan. These CNF clauses then form our $KB_1$ (the KB of the agent providing the explanation). Similar to random knowledge bases, we construct $KB_2$ (the KB of the agent receiving the explanation) by randomly choosing 25% of the clauses from $KB_1$. The formula $\varphi$ that

---

[4]It is one of the domains in the International Planning Competition. See *http://www.plg.inf.uc3m.es/ipc2011-learning/Domains.html*.

we seek to explain is then that no plan of lengths 1 to $h-1$ exists, and that a plan of length $h$ (i.e., the plan found by FASTDOWNWARD) exists. Therefore, combined, that plan must be an optimal plan.

Table 1(b) tabulates our results, where we observe similar trends as in the experiment on random knowledge bases. The key difference is that the runtimes for all four cost functions here are a lot closer to each other, and the reason is because there was only one valid explanation in each problem instance. Thus, regardless of the choice of cost function, that explanation had to be found. Our experiments for larger problems are omitted as they timed out after 6 hours.

## Related Work and Discussions

There is a very large body of work related to the very broad area of explainable AI. We have briefly discussed some of them from the ML literature in Section . We refer readers to surveys by (Adadi and Berrada 2018) and (Dosilovic *et al.* 2018) for more in-depth discussions of this area. We focus below on related work from the KR and planning literature only since we employ KR techniques to solve explainable planning problems in this paper.

**Related Work from the KR Literature:** We note that the notion of an explanation proposed in this paper might appear similar to the notion of a diagnosis that has been studied extensively in the last several decades (e.g., (Reiter 1987)) as both aim at explaining something to an agent. Diagnosis focuses on identifying the reason for the inconsistency of a theory whereas an $m$- or $p$-explanation aims at identifying the support for a formula. The difference lies in that a diagnosis is made with respect to the same theory and $m$- or $p$-explanation is sought for the second theory.

Another earlier research direction that is closely related to the proposed notion of explanation is that of developing explanation capabilities of knowledge-based systems and decision support systems, which resulted in different notions of explanation such as trace, strategic, deep, or reasoning explanations (see review by (Moulin *et al.* 2002) for a discussion of these notions). All of these types of explanations focus on answering why certain rules in a knowledge base are used and how a conclusion is derived. This is not our focus in this paper. The present development differs from earlier proposals in that $m$- or $p$-explanations are identified with the aim of explaining a given formula to a second theory. Furthermore, the notion of an optimal explanation with respect to the second theory is proposed.

There have been attempts to using argumentation for explanation (Cyras *et al.* 2017; Cyras *et al.* 2019) because of the close relation between argumentation and explanation. For example, argumentation was used by (Cyras *et al.* 2019) to answer questions such as why a schedule does (does not) satisfy a criteria (e.g., feasibility, efficiency, etc.); the approach was to develop for each type of inquiry, an abstract argumentation framework (AF) that helps explain the situation by extracting the attacks (non-attacks) from the corresponding AF. Our work differs from these works in that it is more general and does not focus on a specific question.

It is worth to pointing out that the problem of computing a most preferred explanation for $\varphi$ from $KB_1$ to $KB_2$ might look similar to the problem of computing a weakest sufficient condition of $\varphi$ on $KB_1$ under $KB_2$ as described by (Lin 2001). As it turns out, the two notions are quite different. Given that $KB_1 = \{p, q\}$ and $KB_2 = \{p\}$. It is easy to see that $q$ is the unique explanation for $q$ from $KB_1$ to $KB_2$. On the other hand, the weakest sufficient condition of $q$ on $KB_1$ under $KB_2$ is $\perp$ (Proposition 8, (Lin 2001)).

**Related Work from the Planning Literature:** In human-aware planning, the (planning) agent must have knowledge of the human model in order to be able to contemplate the goals of the humans as well as foresee how its plan will be perceived by them. This is of the highest importance in the context of explainable planning since an explanation of a plan cannot be *one-sided* (i.e., it must incorporate the human's beliefs of the planner). In a plan generation process, a planner performs argumentation over a set of different models (Chakraborti *et al.* 2017a); these models usually are the model of the agent incorporating the planner, the model of the human in the loop, the model the agent thinks the human has, the model the human thinks the agent has, and the agent's approximation of the latter.

Therefore, the necessity for plan explanations arises when the model of the agent and the model the human thinks the agent has diverge so that the optimal plans in the agent's model are inexplicable to the human. During a collaborative activity, an explainable planning agent (Fox *et al.* 2017) must be able to account for such model differences and maintain an explanatory dialogue with the human so that both of them agree on the same plan. This forms the nucleus of explanation generation of an explainable planning agent, and is referred to as *model reconciliation* (Chakraborti *et al.* 2017b). In this approach, the agent computes the optimal plan in terms of his model and provides an explanation of that plan in terms of model differences. Essentially, these explanations can be viewed as the agent's attempt to move the human's model to be in agreement with its own. Further, for computing explanations using this approach the following four requirements are considered:

- **Completeness** – No better solution exists. This is achieved by enforcing that the plan being explained is optimal in the updated human model.

- **Conciseness** – Explanations should be easily understandable to the human.

- **Monotonicity** – The remaining model differences cannot change the completeness of an explanation.

- **Computability** – Explanations should be easy to compute (from the agent's perspective).

As our work is motivated by these ideas, we now identify some similarities and connections with our proposed approach. First, it is easy to see that we implicitly enforce the first three requirements when computing an explanation – the notions of completeness and conciseness are captured through the use of our cost functions. We do not claim to satisfy the computability requirement as it is more subjective and is more domain dependent.

In a nutshell, the model reconciliation approach works by providing a model update $\epsilon$ such that the optimal plan is feasible and optimal in the updated model of the human. This is similar to our definition of the explanation generation problem where we want to identify an explanation $\epsilon \subseteq KB_1$ (i.e., a set of formulae) such that $KB_2 \cup \epsilon \models \phi$. In addition, the $\subseteq$-minimal support in Definition 1 is equivalent to *minimally complete explanations* (MCEs) (the shortest explanation). The $\lhd$-general support can be viewed as similar to the *minimally monotonic explanations* (MMEs) (the shortest explanation such that no further model updates invalidate it), with the only difference being that in the general support scenario, the explanations are such that all subsuming $\epsilon$ are also valid supports.

In contrast, *model patch explanations* (MPEs) (includes all the model updates) are trivial explanations and are equivalent to our definition that $KB_1$ itself serves as an $m$-explanation for $KB_2$. Note that, in our approach, we do not allow for explanations on "mistaken" expectations in the human model, as it can be inferred from Proposition 1 (monotonic language $L$). From the model reconciliation perspective, such restriction is relaxed and allowed. However, a similar property can be seen if the mental model is not known and, therefore, by taking an "empty" model as starting point explanations can only add to the human's understanding but not mend mistaken ones.

## Conclusions and Future Work

Explanation generation is an important problem within the larger explainable AI thrust. Existing work on this problem has been done in the context of automated planning domains, where researchers have primarily employed, unsurprisingly, automated planning approaches. In this paper, we approach the problem from the perspective of KR, where we propose a general logic-based framework for explanation generation. We further define two types of explanations, model- and proof-theoretic explanations, and use cost functions to reflect preferences between explanations. Our empirical results with algorithms implemented for propositional logic on both random knowledge bases as well as a planning domain demonstrate the generality of our approach beyond planning problems. Future work includes investigating more complex scenarios, such as one where an agent needs to persuade another that its knowledge base is incorrect.

## Acknowledgment

This research is partially supported by NSF grants 1345232, 1757207, and 1812628. The views and conclusions contained in this document are those of the authors and should not be interpreted as representing the official policies, either expressed or implied, of the sponsoring organizations, agencies, or the U.S. government.

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
