# OpenReview forum: "A General Logic-based Approach for Explanation Generation"
_icaps-conference.org/ICAPS/2019/Workshop/XAIP — XAIP 2019_

### Official Review · AnonReviewer1 · 2019-04-24
**Nice theoretical work on KR-style explanation generation; link to planning and practice a bit tentative**

**Rating:** 3
**Confidence:** 2

**Review:**

This is a nicely conducted formal piece of research on explanation generation from a KR point of view, where the explanation consists, given KB_1 and KB_2 where only KB_1 entails phi, in a subset epsilon of KB_1 adding which to KB_2 results in entailment of phi. The paper spells this out formally, and conducts experiments measuring explanation size and runtime on random formulas as well as formulas taken from a Blocksworld instance.

Overall this is nice work. The link to practice is tentative as it is not clear (nor even discussed) how the formulas epsilon will be presented to users, and under what conditions this is feasible. If the authors could add some thoughts in this direction, that would be appreciated.

The link to planning is tentative as the paper is about KR. As a matter of fact, the paper is being motivated by its distinction from planning, going "beyond" previous approaches by considering not only planning. This actually feels a but funny, as  a submission to an explainable *planning* workshop. Perhaps the authors should have taken care to reformulate their IJCAI submission (if that is what this is) a bit more, for this submission?

In any case, a link to planning is there through the planning-as-SAT approach, materialized in the paper by the experiment on Blocksworld. There, phi is the property that no plan exists up to optimal plan length -1, but a  plan exists at that length; KB_1 is a standard SAT encoding; KB_2 is 25% random clause seubset of KB_1. I can't say I find this setup particularly meaningful -- random clauses?? -- but I do find that the setup per se, using this approach on planning-as-SAT, is meaningful.

So overall I lean to accept. Perhaps, as this is likely to be of interest to a more specialized audience, a poster presentation would be more appropriate than a long technical talk. And I do advise the authors to refomulate their motivation going beyond planning.

On that note: It is actually not clear to me why the authors choose to motivate their work this way. In my perception, it is not like planning is at the forefront of current XAI research. It seems to me a much simpler pitch for this paper would have been to take the KR perspective, mentioning the XAI work in ML, saying that model-based approaches are potentially more suited for XAI, saying that KR is a core field in model-based AI, then outlining what's new here relative to previous KR work. The work on XAIP is then merely an aside that can be easily deferred to the related work section.

Disclaimer: I am not a KR expert and am assuming/trusting the authors that there is sufficient novelty here over previous KR works.

Minor: There are several broken refs in the related work section. Please fix.

---

### Official Review · AnonReviewer6 · 2019-05-14
**A model reconciliation equivalent in the context of knowledge bases**

**Rating:** 3
**Confidence:** 2

**Review:**

The paper provides an interesting perspective on explanations between two knowledge bases, and runs parallel to the work on model reconciliation in the planning literature.

1) I am slightly disappointed that the authors did not make an effort to make connections to planning concepts. After all, its an XAI"P" workshop. The example in BlocksWorld is nice, but really comes at the end as an afterthought.

It would be very helpful if both model- and proof- theoretic explanations could be scoped in planning terms in the beginning. Like "no plan of lengths 1 to h-1 exists" and "a plan of length h exists". What kind of questions is model-theoretic explanation solving? What about proof-theoretic?

2) Many of the concepts here have parallels to model reconciliation in planning. It would be great to see a discussion on those parallels. For example,

-- minimality of explanations: Minimally complete explanations (MCEs) in [Chakraborti et al. IJCAI 2017] seem to be equivalent to subset-minimality in Definition 1.
-- monotonicity of explanations: similarly, minimally monotonic explanations (MMEs) talk about the shortest explanation that can be given such that no further model updates invalidate it. This is somewhat complimentary to the general-support scene: except that these are explanations such that all subsuming eplisons are also valid supports. On the other hand, model patch explanations (MPEs) are trivial explanations as indicated by the fact that "KB1 is  itself an m-explanation for c from KB1 for KB 2".
-- negative clauses + monotonicity of logic (Proposition 1) and empty models: This is an interesting artifact, the implication being that explanation can address missing or lack of understanding of the user, but not mistaken expectations (in monotonic L). From the model reconciliation point of view, there is no such restriction. However, if the mental model is not known, and the we take the starting point as an "empty model" then one can see a similar property that explanations can only add to the user's understanding but not fix mistaken ones.
-- preferences: Explanations in planning under the model reconciliation framework are also non-unique and may have arbitrary preferences from the user's point of view. The use of non-monotonic cost functions is interesting. I wonder whether there might be ways of mapping them to revealed preferences of users among otherwise "logically" equivalent explanations. This might be of interest: https://ieeexplore.ieee.org/document/8673097

I had a lot of fun making these connections to the explanation literature in planning. I would request that the authors try to make a similar XAIP argument to the paper -- especially Question 1 above.

Other:
-- Definition 3: K --> L?
-- ?s in final section

---

> ### Author Response · Authors · 2019-05-15
> **Response to comments**
>
> 1) The reason that we didn’t make any explicit connections to planning concepts is because we wanted to maintain the generality of our proposed framework. Casting the model- and proof-theoretic explanations to planning terms is indeed a very interesting idea. Some of the questions they could potentially solve might be “why is this the optimal plan?” or “why is a particular action needed?”, etc.
>
> 2) Thank you for taking the time in making connections with the explanation literature in planning. Indeed, there are many similarities with the work of [Chakraborti et al. IJCAI 2017]. As a matter of fact, we were highly motivated by this work. We aim to focus on XAIP and address all these similarities in a next iteration of our paper.

---

### Official Review · AnonReviewer2 · 2019-05-14
**Explanations the KR context.**

**Rating:** 2
**Confidence:** 2

**Review:**


The authors address the problem of agent explanations, where an agent has to explain its bahaviour to another agents. The problem is formalised in the context of Knowledge Representation and Reasoning.
Here, an agent with knowledge base (KB1) that satisfies \phi has to explain to another agent (user) - having a KB2 that does not satisfy \phi - how KB1 can be trasformed to KB2, thus considering such a transformation \epsilon as an explanation.

As far as I know, the problem resembles to the Ontology Alignment, that is the process to determine correspondences between ontologies (i.e., the \epsilon that explain how two distinct KBs can fit).
The paper is definitively interesting, though the approach that the authors use limits the impact and the relevance of the paper in the XAIP workshop, and this is my main concern to the acceptance.

COMMENTS:

-- The experimental results (approach applied on Block World using some classical planners) do not are unclear to me, as they do not discuss pros/cons/limits of one approach wrt to another. Furthermore, I'm not convinced that the cost function can be used as a qualitative metric for selecting the best explanation (it is a quantitative metric instead).
-- I appreciate the comparison of the approach with AI Planning, though I was expecting to see the effectiveness of the proposed approach against ML techniques, that are used for performing such tasks.


MINOR:
-- In related works, there are some missing citations (?)
-- There are some missing references to sections

---

### Public Comment · ~Alexey_Ignatiev1 · 2019-05-17
**Paper misses related work.**

Although this paper cites a few surveys, e.g. (Guidotti et al.2018), given the topic covered in this paper, it would be important to compare with published related work (on logic-based approaches to computing explanations) that is overlooked in the submitted version, namely:

Andy Shih, Arthur Choi, Adnan Darwiche: A Symbolic Approach to Explaining Bayesian Network Classifiers. IJCAI 2018.

Alexey Ignatiev, Nina Narodytska, João Marques-Silva: Constraint-Based Explanations of Machine Learning Models. AAAI 2019.

---

### Decision · Program_Chairs · 2019-05-15

**Decision:**

Accept

**Comment:**

While the reviewers view this paper somewhat critically, in the spirit of making the workshop a venue for discussion and feedback we decided to reject only those papers with strong reject votes.

Please address the review criticism as best possible for the final paper version and its presentation at the workshop. In particular, please carefully discuss the relation to/links to XAIP, and the XAIP literature. Looking forward to discuss your work at the workshop!